# Combining Breast Cancer Risk Prediction Models

**DOI:** 10.3390/cancers15041090

**Published:** 2023-02-08

**Authors:** Zoe Guan, Theodore Huang, Anne Marie McCarthy, Kevin Hughes, Alan Semine, Hajime Uno, Lorenzo Trippa, Giovanni Parmigiani, Danielle Braun

**Affiliations:** 1Department of Epidemiology and Biostatistics, Memorial Sloan Kettering Cancer Center, New York, NY 10017, USA; 2Vertex Pharmaceuticals, Boston, MA 02210, USA; 3Department of Biostatistics, Epidemiology and Informatics, University of Pennsylvania, Philadelphia, PA 19104, USA; 4Department of Surgery, Medical University of South Carolina, Charleston, SC 29425, USA; 5Advanced Image Enhancement, Fall River, MA 02720, USA; 6Department of Data Science, Dana-Farber Cancer Institute, Boston, MA 02115, USA; 7Department of Medicine, Harvard Medical School, Boston, MA 02115, USA; 8Department of Biostatistics, Harvard T.H. Chan School of Public Health, Boston, MA 02115, USA

**Keywords:** BRCAPRO, BCRAT, model aggregation, ensemble learning, stacking

## Abstract

**Simple Summary:**

BRCAPRO is a widely used breast cancer risk prediction model based on family history. A major limitation of this model is that it does not consider non-genetic risk factors. We expand BRCAPRO by combining it with another popular model, BCRAT, that uses mostly non-genetic risk factors, and show that the expanded model can achieve improvements in prediction accuracy over both BRCAPRO and BCRAT.

**Abstract:**

Accurate risk stratification is key to reducing cancer morbidity through targeted screening and preventative interventions. Multiple breast cancer risk prediction models are used in clinical practice, and often provide a range of different predictions for the same patient. Integrating information from different models may improve the accuracy of predictions, which would be valuable for both clinicians and patients. BRCAPRO is a widely used model that predicts breast cancer risk based on detailed family history information. A major limitation of this model is that it does not consider non-genetic risk factors. To address this limitation, we expand BRCAPRO by combining it with another popular existing model, BCRAT (i.e., Gail), which uses a largely complementary set of risk factors, most of them non-genetic. We consider two approaches for combining BRCAPRO and BCRAT: (1) modifying the penetrance (age-specific probability of developing cancer given genotype) functions in BRCAPRO using relative hazard estimates from BCRAT, and (2) training an ensemble model that takes BRCAPRO and BCRAT predictions as input. Using both simulated data and data from Newton-Wellesley Hospital and the Cancer Genetics Network, we show that the combination models are able to achieve performance gains over both BRCAPRO and BCRAT. In the Cancer Genetics Network cohort, we show that the proposed BRCAPRO + BCRAT penetrance modification model performs comparably to IBIS, an existing model that combines detailed family history with non-genetic risk factors.

## 1. Introduction

Breast cancer is the second most common cancer and the second leading cause of cancer death in women in the U.S. [1,2]. Identifying individuals at high risk is critical for guiding decisions about risk management and prevention, including screening, genetic counseling and testing, and preventative procedures. At least 24 breast cancer risk prediction models have been developed for clinical use [3]. These models estimate either an individual’s risk of carrying a pathogenic mutation in a breast cancer susceptibility gene, an individual’s future risk of breast cancer, or both. They are based on a wide range of risk factors, methodologies, and study populations. Regression-based models, such as the Breast Cancer Risk Assessment Tool (BCRAT, also known as the Gail model) [4,5,6,7], use hormonal/reproductive risk factors (such as age at first live birth), history of benign disease, and simple summaries of family history. Other models, such as BRCAPRO [8], BOADICEA [9,10,11], and IBIS [12], are Mendelian models that use detailed family history information and principles of genetic inheritance. IBIS and BOADICEA [11] take into account hormonal/reproductive risk factors and history of benign disease. BRCAPRO currently does not use non-genetic risk factors, which limits its predictive accuracy [13,14]. Because it is widely adopted in clinical practice [15], adding non-genetic risk factors to the model is crucial. We address this problem by combining it with the widely used BCRAT model, which is based on a largely complementary set of risk factors, most of which are non-genetic.

BRCAPRO is a family history-based model that provides carrier probabilities for breast cancer susceptibility genes BRCA1 and BRCA2 as well as future risk estimates for invasive breast and ovarian cancer. It translates family history data into risk estimates using Mendelian laws of inheritance, Bayes’ rule, and literature-based estimates of mutation prevalence and penetrance (age-specific probability of developing cancer given genotype). In contrast, BCRAT estimates an individual’s future risk of invasive breast cancer based on a relative hazard model that includes age, hormonal and reproductive risk factors, history of benign disease, and first-degree family history of breast cancer. The relative hazard model was originally trained using case-control data from Caucasian women participating in a U.S. mammography screening program. It was later updated for African-American [5], Asian-American [6], and Hispanic [7] women. Although there is overlap in the inputs to BRCAPRO and BCRAT, the two models are largely complementary (Figure 1). BRCAPRO uses potentially extensive family history information, while BCRAT considers only first-degree female relatives with breast cancer. BCRAT considers several non-genetic risk factors that are not considered by BRCAPRO, including age at menarche, age at first live birth, and number of breast biopsies. A validation study in a large U.S. screening cohort found that 6-year risk predictions from BRCAPRO and BCRAT had a moderate correlation of 0.53 [16]. Because BRCAPRO and BCRAT embed different information, combining these models can potentially lead to accuracy gains.

There already exist models that incorporate both detailed family history information and non-genetic risk factors, namely, IBIS (also known as the Tyrer–Cuzick model) [12] and BOADICEA [11]. These two models have outperformed BRCAPRO and BCRAT in two previous validation studies [13,14]. However, we believe it is valuable to investigate the combination of BRCAPRO and BCRAT in order to determine (1) how much predictive value non-genetic risk factors add to BRCAPRO, (2) how much predictive value detailed family history adds to BCRAT, and (3) whether model combination can achieve competitive performance compared to developing a hybrid model from the ground up. Moreover, BRCAPRO and BCRAT continue to be widely used in clinical practice. A model that directly combines and improves upon predictions from BRCAPRO and BCRAT may be appealing to current users of these two models.

We consider two combination approaches: (1) penetrance modification and (2) ensemble learning. While we focus on BRCAPRO and BCRAT in this paper, these approaches can be applied in general to expand any Mendelian model to include additional risk factors. The first approach modifies the penetrance functions in BRCAPRO to account for the effects of the BCRAT risk factors. We develop a penetrance modification model, BRCAPRO + BCRAT (M), using a relative hazard approach that has similarities to the one used in IBIS.

Ensemble learning consists of training multiple base models and combining their predictions. A wide variety of ensemble methods have been developed, including stacking [17], which involves training a meta-model to optimally combine predictions from the base models, bagging [18,19], which involves averaging models trained on bootstrap samples of the original data, and boosting [20], which involves constructing an ensemble by sequentially adding new models that are trained to correct the errors of previous ones. The rationale for these methods is that combining predictions can reduce variance and expand the set of functions that can be represented by any individual base model [21]. Extensive research, both empirical [21,22] and theoretical [23,24,25,26,27], has shown that ensembles can achieve performance gains over their base models, especially when the base models produce dissimilar predictions [28,29]. Debray et al. (2014) [30] demonstrated the value of aggregating published prediction models using real and simulated data on deep venous thrombosis and traumatic brain injury. Across various scenarios, model averaging and stacking outperformed model recalibration [31,32] and performed as well as or better than developing a new model from the ground up. Moreover, the authors noted that stacking is more efficient than model averaging, as stacking has fewer unknown parameters. In the setting of breast cancer risk prediction, Ming et al. (2019) [33] showed that boosting and random forests, a form of bagging, were able to achieve higher discriminatory accuracy than BCRAT and BOADICEA. In this paper, we develop a stacked logistic regression ensemble, BRCAPRO + BCRAT (E), that takes predictions from BRCAPRO and BCRAT as input.

We compare the performance of the combination models to the individual BRCAPRO and BCRAT models in simulations and a data application, where we use training data from the Newton-Wellesley Hospital (NWH) and validation data from the Cancer Genetics Network (CGN). In the data application, we use IBIS as a reference for comparison in order to evaluate the strategy of developing a hybrid model from the ground up versus the strategy of combining existing models.

## 2. Materials and Methods

### 2.1. Problem Definition

Given a female individual without a previous diagnosis of breast cancer who presents for risk assessment (the counselee), the goal is to predict her risk of developing invasive breast cancer within τ years (where τ is a pre-specified positive integer) based on family history and other risk factors while accounting for death from other causes as a competing risk. Below, we provide an overview of the risk prediction models compared in this paper. A more detailed description of the models with formal notation is provided in Appendix A.

### 2.2. Existing Models

#### 2.2.1. BRCAPRO

BRCAPRO [8] estimates the probability of carrying a deleterious germline mutation in BRCA1 and BRCA2 using Bayes’ rule, laws of Mendelian inheritance, mutation prevalence and penetrance, and family history of breast and ovarian cancer. In addition, it estimates future risk of breast and ovarian cancer based on the carrier probabilities and penetrances. The family history information used by BRCAPRO includes the following variables for each family member (missing values are allowed): gender, breast cancer status, age of onset of breast cancer if applicable, ovarian cancer status, age of onset of ovarian cancer if applicable, and current age or age at death. Information on preventative interventions (mastectomy/oophorectomy) and genetic testing results for BRCA1/BRCA2 can also be included.

BRCAPRO calculates the counselee’s probability of having a given genotype using Bayes’ rule and Mendelian laws of inheritance. This calculation depends on mutation prevalence and penetrance estimates obtained from published studies (see Appendix A for more details). The prevalences are ethnicity-specific (in particular, different prevalences are used for Ashkenazi Jewish and non-Ashkenazi Jewish families), and the penetrances, which are functions of age, are cancer- and sex-specific. The penetrance functions for non-carriers are based on rates from the Surveillance, Epidemiology, and End Results (SEER) program, and are race-specific, while the penetrance functions for carriers are from a meta-analysis of published studies [34]. After estimating the carrier probabilities, BRCAPRO calculates future risk of breast and ovarian cancer through a weighted average of the genotype-specific risks.

We ran BRCAPRO using the BayesMendel R package [35] version 2.1-6.1, selecting the crude risk option.

#### 2.2.2. BCRAT

BCRAT [4,5,6,7] estimates the relative hazard of developing breast cancer based on age and the following risk factors: age at menarche, number of benign breast biopsies, age at first live birth, number of female first-degree relatives with breast cancer, and history of atypical hyperplasia. The relative hazard model includes interactions between age and number of biopsies as well as age at first live birth and number of affected relatives. The regression coefficients were estimated from U.S. case-control studies. Separate models were fit to data from White, African-American, Asian, and Hispanic women to obtain race/ethnicity-specific estimates. In the future risk calculation, the race-specific baseline hazard of breast cancer (based on SEER data) is multiplied by the relative hazard associated with the risk factors (see Appendix A).

We ran BCRAT using the BCRA R package (https://cran.r-project.org/web/packages/BCRA/index.html, accessed on 20 March 2020), version 2.1.

#### 2.2.3. IBIS

In our data application, we examine the IBIS model [12,36] as a reference for comparison, as it combines detailed family history information with non-genetic risk factors. It first calculates carrier probabilities and risk of breast cancer based on family history, then incorporates additional risk factors (age at menarche, age at menopause, height, body mass index, age at first live birth, menopausal hormone therapy, atypical hyperplasia, lobular carcinoma in situ, breast density, and a polygenic risk score for breast cancer) via a relative hazard model. The carrier probabilities are calculated using a similar approach as in BRCAPRO; however, in addition to BRCA1 and BRCA2, IBIS considers a hypothetical low-penetrance susceptibility gene that acts as a surrogate for other unspecified breast cancer susceptibility genes. Estimates of the prevalence and penetrance of BRCA1 and BRCA2 were obtained from previously published studies, while the prevalence and penetrance of the hypothetical gene were estimated from a Swedish population-based study. The penetrance function for non-carriers is based on incidence rates from the Thames Cancer Registry.

IBIS calculates a weighted average of the cumulative penetrances for each genotype to obtain the risk of breast cancer conditional on family history only. In the final future risk calculation, IBIS multiplies the hazard of breast cancer conditional on family history by the relative hazard associated with the additional risk factors (see Appendix A for details).

Software for running IBIS is available at http://www.ems-trials.org/riskevaluator/ (accessed on 20 April 2018). We used the command line program for version 8, selecting the competing mortality option.

### 2.3. Model Combination Approaches

#### 2.3.1. Penetrance Modification Model: BRCAPRO + BCRAT (M)

The penetrance modification approach combines BRCAPRO and BCRAT by using the BCRAT relative hazard to modify the calculation of the non-carrier future risk in BRCAPRO. Specifically, the baseline non-carrier hazard in the BRCAPRO future risk calculation is scaled by the BCRAT relative hazard based on a discrete proportional hazards model (see Appendix A for details). In the current implementation, we apply the modification to only the non-carrier future risk calculation, as BCRAT is not recommended for known carriers of BRCA1/2 mutations [37]. As in BRCAPRO, the final risk is a weighted average of the genotype-specific risks.

We refer to this model as the penetrance modification model, BRCAPRO + BCRAT (M), because the modification of the hazard function induces a modification of the corresponding penetrance function. This combination approach is similar to replacing the non-carrier future risk from BRCAPRO with the future risk from BCRAT. However, it is not identical because BRCAPRO and BCRAT use slightly different baseline hazards (see Figure A1 in Appendix B for plots of the hazards used in BCRAT and BRCAPRO).

The relative hazard approach for incorporating the BCRAT risk factors has similarities to the approach used by IBIS, except that IBIS averages the genotype-specific risks before incorporating non-genetic risk factors while BRCAPRO + BCRAT (M) incorporates the BCRAT risk factors before averaging the genotype-specific risks. The advantage of the latter approach is that it allows the effects of the BCRAT risk factors to differ by genotype. Genotype-specific effects have been observed for certain BCRAT risk factors, such as age at menarche (see [38] for a review). However, in general, the effects of the BCRAT risk factors on carriers are not well-studied; only a limited number of prospective studies have been done and they have had small sample sizes [38]. Therefore, the current version of BRCAPRO + BCRAT (M) modifies only the non-carrier hazard.

#### 2.3.2. Ensemble Model: BRCAPRO + BCRAT (E)

The second model combination approach involves training a stacked ensemble model [17] that uses BRCAPRO and BCRAT as the (pre-trained) base models. We consider a logistic regression ensemble that predicts τ-year risk of breast cancer for fixed τ, as well as a time-to-event ensemble that provides predictions for different time intervals.

As in Debray et al. (2014) [30], for a fixed value of τ we can combine the τ-year BRCAPRO and BCRAT predictions using a logistic regression model that includes the two predictions as covariates along with an interaction term (their product). Other covariates and/or published models can be included as inputs. We refer to this model as BRCAPRO + BCRAT (E).

Alternatively, we can use BRCAPRO and BCRAT predictions and their product as covariates in a time-to-event model. We consider a Fine-Gray proportional subdistribution hazards model for breast cancer, accounting for death as a competing risk (see Appendix A for details). The BRCAPRO and BCRAT predictions are calculated at baseline for a predefined time point $\tau^{*}$ and do not vary with $\tau^{*}$. We refer to this model as BRCAPRO + BCRAT (E2).

In contrast to the penetrance modification model, which can be implemented using published parameter estimates, the ensemble models require training. If predictions from BRCAPRO and BCRAT are highly correlated in the training data, then multicollinearity can lead to unstable coefficient estimates [39]. Moreover, the ensemble models might perform poorly in external validation if the training data are not representative of the validation data. Ideally, they should be trained using a prospective cohort that is representative of the target population. Otherwise, recalibration or reweighting methods can be used to account for differences in the covariate distributions between the training population and the target population. One widely used method is importance weighting [40], which weights each training observation by the ratio of the joint probability distributions of the covariates in the target and training populations [40]. The importance weights can be estimated using kernel mean matching [41], Kullback–Leibler importance estimation [42], or least squares importance fitting [43].

In the simulations and data application, we applied a square root transformation to the BRCAPRO and BCRAT predictions prior to fitting the ensemble models, as the distributions of the predictions were highly right-skewed. For BRCAPRO + BCRAT (E2), we used $\tau^{*}=5$ to compute the covariates.

### 2.4. Model Evaluation Metrics

In the simulations and data application, we considered the binary outcome of being diagnosed with breast cancer within τ=5 years. In the data application, we additionally considered the time-to-event outcome over the course of follow-up, as many of the counselees in the validation dataset were followed for more than five years; however, there was substantial variability in follow-up times.

We used five performance measures [44]: (1) the ratio of observed (O) to expected (E) events, where E is calculated by summing everyone’s predicted probabilities (a measure of calibration, with 1 indicating perfect calibration); (2) the area under the receiver operating characteristic curve (AUC) or concordance (C) statistic, which is the probability that an individual who experiences the event has a higher score than an individual who does not, and is a measure of discrimination; (3) the Brier score, which is the mean squared difference between the predicted probabilities and actual outcomes; (4) the standardized net benefit (SNB) [45,46,47], which is the difference between the true positive rate and a weighted false positive rate, where the weight is based on a pre-specified threshold for classifying individuals as high-risk versus low-risk (the weight is the ratio of the odds of the threshold risk to the odds of the outcome); and (5) the logarithmic score [48], which is the negative log-likelihood. Calibration was assessed using both overall O/E and calibration plots of O/E by risk decile. We calculated SNB for only the binary outcome, using a 5-year risk threshold of 1.67% (the clinical 5-year risk threshold for eligibility for chemoprevention). We report the Brier score and logarithmic score in terms of relative difference with respect to BRCAPRO, as these metrics are prevalence-dependent, and as such are more difficult to interpret on their original scale.

In the data application, certain individuals in the validation dataset were censored before τ=5 years. To account for censoring, we used inverse probability of censoring weights (IPCW) [49,50] to calculate the O/E, AUC, Brier score, and logarithmic score for the binary outcome; individuals with observed outcomes were used to calculate the performance measures and weighted by their inverse probability of not being censored by the minimum of (1) the end of the 5-year projection period and (2) the time at which they were diagnosed with breast cancer. Individuals who were censored were not directly used to calculate the performance measures, and were instead used to estimate the censoring distribution via the Kaplan–Meier estimator.

For the time-to-event outcome, O/E was calculated by comparing the numbers of observed and expected cases across the entire study period; for the expected number of cases E, we predicted risk up to the end of the individual follow-up time for each counselee, meaning that τ varied across counselees. In addition, we used time-to-event versions of the C-statistic and logarithmic score. The time-to-event C-statistic [51] is the probability of an individual with a shorter time-to-event having a higher score than an individual with a longer time-to-event. A fixed prediction period of τ=10 years was used to calculate the C-statistic (which requires the same prediction period for everyone), as ten years was the maximum event time observed in the data. We used a version of the logarithmic score that accounted for competing risks [52]. In the absence of censoring, the logarithmic score is a strictly proper scoring rule for the problem of predicting a probability distribution [53,54]. We have previously shown that the competing risks version remains strictly proper under non-informative censoring [52].

We calculated 95% bootstrap confidence intervals (CIs) for all of the performance measures except for the time-to-event C-statistic, for which a 95% CI was obtained using perturbation resampling [51]. For each performance measure, we looked at pairwise comparisons of the models across bootstrap replicates of the validation set.

### 2.5. Simulations

We performed simulations to compare the 5-year performance of the combination models to that of the individual BRCAPRO and BCRAT models in a setting where the assumptions of the penetrance modification model hold.

First, we generated each counselee’s baseline family history, consisting of (1) the family structure (i.e., the number of sisters, number of brothers, etc.), (2) dates of birth, (3) genotypes, (4) cancer ages, and (5) death ages.

We simulated pedigrees in order to mimic the family structures observed in real families from the CGN dataset (the validation dataset for the data application, described in Section 2.6.2). We restricted the family members to first- and second-degree relatives of the counselee.

For counselees, dates of birth and baseline dates for risk assessment were sampled from the CGN dataset. For non-counselees, dates of birth were generated relative to the counselee’s date of birth by assuming that the age difference between a parent and a child had a mean of 27 and standard deviation of 6. We generated the birth dates of the counselee’s parents and children based on the counselee’s birth date, then the birth dates of the counselee’s grandparents and siblings based on the birth dates of the parents, then the birth dates of the counselee’s aunts and uncles based on the birth dates of the grandmothers.

Next, we generated the BRCA1/2 genotypes for each family member. We first generated the genotypes of the grandparents (founders) using the default Ashkenazi Jewish allele frequencies for BRCA1 and BRCA2 in BRCAPRO, which reflect a higher-risk population (CGN participants represent a higher-risk population than the general population, as they were selected for family history of cancer). For individuals in subsequent generations, we generated genotypes according to Mendelian inheritance.

For each individual, we generated baseline breast and ovarian cancer phenotypes conditional on genotype. Age of onset was sampled from {1, 2, …, baseline age}, with probabilities provided by the genotype-specific penetrance functions from BRCAPRO. The probability of being unaffected at baseline was provided by one minus the cumulative penetrance up to the baseline age. Counselees were assumed to be alive at baseline, while for each non-counselee we generated a death age from a distribution with mean of 80 and standard deviation of 15. If an individual had an age of onset greater than their age at death, then the individual’s cancer status was changed to unaffected. Counselees with breast cancer at baseline were excluded from the analyses.

We then generated baseline BCRAT covariates (other than number of affected first-degree relatives and age at first live birth, which were calculated from the baseline family history) by sampling values from the distribution in the CGN. Values for different covariates were sampled independently of each other. The BCRAT covariates were used to modify the BRCAPRO non-carrier penetrance to obtain the BRCAPRO + BCRAT (M) non-carrier penetrance (Equation (Equation 16)).

For counselees who did not have breast cancer at baseline, future ages of onset were generated from the BRCAPRO + BCRAT (M) penetrances (for carriers, the BRCAPRO + BCRAT (M) penetrances are the same as in BRCAPRO), which were rescaled to be conditional on not having developed cancer by the baseline age. Cases were defined as counselees who developed breast cancer within five years of their baseline age. The 5-year outcomes were not subject to censoring.

We simulated 100,000 families in total. After excluding 4443 counselees with breast cancer at baseline, we used the first randomly generated 50,000 counselees to train the ensemble model (similar to the size of the training set in the data application; see Section 2.6.1) and the remaining 45,557 for validation. There were 814 cases in the training set and 724 cases in the validation set.

### 2.6. Data Application

Using data from Newton-Wellesley Hospital (NWH), we trained ensemble models for the binary outcome at τ=5 (Model (Equation 19)) and for the time-to-event outcome (Model (Equation 20)). We validated these models, along with BRCAPRO + BCRAT (M), BRCAPRO, BCRAT, and IBIS, using data from the CGN. We assessed performance based on both the binary and time-to-event outcomes. In addition, we looked at performance stratified by family history, with strata defined based on the NCCN criteria for further genetic risk evaluation [55].

In the analyses, we excluded women with any of the following conditions prior to baseline: invasive breast cancer, ductal carcinoma in situ, lobular carcinoma in situ, bilateral mastectomy or bilateral oophorectomy. In addition, we excluded women who tested positive for BRCA1/2 prior to baseline (BCRAT requirement), women <20 years old at baseline (BCRAT requirement), and women with a projection interval extending beyond 85 years of age (IBIS requirement).

The characteristics of the training and validation datasets are described below and summarized in Table 1.

#### 2.6.1. Training Dataset (NWH)

After applying the exclusion criteria, the training cohort consisted of 37,881 women who visited the breast imaging department of NWH in Newton, Massachusetts for screening or diagnostic imaging from February 2007 through December 2009. During the initial (baseline) visit, information was collected on personal and family history of cancer, reproductive history, sociodemographic factors, and lifestyle factors. Family history was limited to relatives with cancer. Breast cancer diagnoses through 2015 were determined from the Massachusetts State Cancer Registry, Partners Hospital Cancer Registries, and patient self-reporting. The median age of the counselees was 49, with an interquartile range (IQR) of 43–58; 30,758 (81.2%) of the counselees were White, while 5684 (15.0%) had at least one affected first- or second-degree relative. The median follow-up time was 6.7 years (IQR 6.3–7.2). All counselees were followed for at least six years. There were 495 counselees (1.3%) who developed breast cancer within five years of baseline and 714 counselees (1.9%) who developed breast cancer over the course of follow-up.

Because the NWH cohort represents a general screening population and the CGN validation cohort (described below) represents a higher-risk population enriched for family history of cancer, we applied importance weights to the training data based on the distributions of the BCRAT covariates, 5-year BCRAT predictions, and 5-year BRCAPRO predictions when we fit the ensemble models. The weights were estimated using least squares importance fitting via the densratio R package (https://cran.r-project.org/web/packages/densratio/index.html (accessed on 20 April 2018)).

#### 2.6.2. Validation Dataset (CGN)

The validation cohort consisted of 7314 women who enrolled in the CGN, a national research network of fifteen academic medical centers that was established for the purpose of studying inherited predisposition to cancer. Enrollment began in 1999 and ended in 2010. One of the criteria for enrollment was a personal and/or family history of cancer. Participants provided information on personal and family history of cancer, sociodemographic factors, and lifestyle factors through an initial (baseline) phone interview and annual follow-up updates. From 2009 onward, information was collected on reproductive history, cancer treatments, cancer screening results, and genetic testing results.

The median age of the counselees was 47 (IQR 38–57); 6104 (83.5%) of the counselees were White, while 3143 (42.9%) had at least one female first-degree relative with breast cancer. The median follow-up time was 7.3 years (IQR 6.0–8.3) and 934 (12.8%) counselees were censored within five years of baseline without being diagnosed with breast cancer. Of the counselees, 159 (2.2%) developed breast cancer during follow-up, with 112 of the diagnoses occurring within five years of baseline. Demographic characteristics stratified by center are provided in Table A4 of Appendix D. Because follow-up times and breast cancer incidence rates varied by center, we estimated the censoring distribution (which was needed to calculate several of the performance measures) by separately fitting a Kaplan–Meier curve for each center.

Information on certain risk factors was missing or incomplete. We did not have information on atypical hyperplasia (used in BCRAT and IBIS), breast density (used in IBIS), polygenic risk scores (used in IBIS), or hormone replacement therapy (used in IBIS). Participants were asked whether they had ever had a benign breast biopsy, but were not asked about the number of biopsies (categorized as 0, 1, or ≥2 in BCRAT). Because participants were asked about reproductive history starting in 2009, 4157 (56.8%) were missing age at menarche (used in BCRAT and IBIS). Information on Ashkenazi Jewish heritage (used in BRCAPRO and IBIS) was not available for the University of Washington (UWASH) center. We coded the missing variables according to the specifications of the software for each model. Number of breast biopsies was coded as 1 for participants who indicated that they had previously had a biopsy.

## 3. Results

### 3.1. Simulation Results

The performance measures are shown in Table 2 and calibration plots are shown in Figure 2. We chose not to report the performance of the IBIS model in this section, as the data generating model is more favorable to BRCAPRO and BCRAT. BRCAPRO + BCRAT (M), the true model, had the best performance, although the ensemble models were able to achieve similar performance to the true model and performance gains over BRCAPRO and BCRAT. The combination models were well-calibrated overall, with O/E = 1.01 (95% CI 0.94–1.08) for BRCAPRO + BCRAT (M), O/E = 0.98 (95% CI 0.91–1.04) for BRCAPRO + BCRAT (E), and O/E = 0.99 (95% CI 0.92–1.05) for BRCAPRO + BCRAT (E2). BRCAPRO and BCRAT underpredicted the number of cases, with O/E = 1.15 (95% CI 1.07–1.23) for BRCAPRO and O/E = 1.14 (95% CI 1.05–1.21) for BCRAT. The combination models were well-calibrated in each decile of risk (Figure 2), while BRCAPRO and BCRAT were more prone to underpredicting risk in certain deciles. The combination models had slightly higher AUCs than BRCAPRO and BCRAT: 0.69 (95% CI 0.67–0.71 for BRCAPRO + BCRAT (M), 0.68 (95% CI 0.67–0.70) for BRCAPRO + BCRAT (E) and BRCAPRO + BCRAT (E2), 0.67 (95% CI 0.65–0.69) for BRCAPRO, and 0.66 (95% CI 0.64–0.68) for BCRAT. In addition, the combination models performed better than BRCAPRO and BCRAT with respect to the Brier score, logarithmic score, and SNB. Across 1000 bootstrap replicates of the validation dataset, BRCAPRO + BCRAT (M) outperformed BRCAPRO and BCRAT with respect to all performance measures in more than 95% of the replicates and the ensemble models outperformed BRCAPRO and BCRAT with respect to all performance measures except O/E in more than 95% of the replicates. The ensemble models had better O/E ratios than BRCAPRO and BCRAT more than 92% of the time. BRCAPRO + BCRAT (M) outperformed BRCAPRO + BCRAT (E) and BRCAPRO + BCRAT (E2) more than 97% of the time with respect to each of AUC, Brier score, and logarithmic score.

### 3.2. Data Application Results

#### 3.2.1. Five-Year Binary Outcome

The performance measures based on the five-year outcome are shown in Table 3 (overall performance) and Table 4 (performance stratified by family history). Figure 3 shows the distributions of the predictions as well as pairwise correlations between models, while Figure 4 shows the calibration plots. The weights from the ensemble models are provided in Appendix C.

As seen in Figure 3, the lowest correlations were observed between models with smaller overlaps of the input variables, as expected. BRCAPRO and BCRAT had the lowest correlation (ρ=0.44). Even models with higher correlations displayed divergent predictions. Correlations of predictions from BRCAPRO + BCRAT (M) with predictions from each of the other models ranged from ρ=0.78 with BRCAPRO to ρ=0.93 with BRCAPRO + BCRAT (E). BRCAPRO + BCRAT (E) and BRCAPRO + BCRAT (E2), which assigned a higher weight to BCRAT than to BRCAPRO (Appendix C), were more highly correlated with BCRAT (ρ=0.93) than with BRCAPRO ((ρ=0.67 for (E) and ρ=0.62 for (E2)).

On aggregate, BRCAPRO + BCRAT (M) (O/E = 1.03, 95% CI 0.84–1.23) and IBIS (O/E = 0.98, 95% CI 0.80–1.17) were well-calibrated, while BCRAT (O/E = 1.15, 95% CI 0.94–1.37), BRCAPRO + BCRAT (E) (O/E = 1.17, 95% CI 0.96–1.40), BRCAPRO + BCRAT (E2) (O/E = 1.20, 95% CI 0.98–1.43), and BRCAPRO (O/E = 1.30, 95% CI 1.07–1.56) underestimated risk. When calibration was considered by decile (Figure 4), 8 out of 60 intervals overall failed to cross the diagonal, compared to the result of 3 expected by chance when all models are well-calibrated. BRCAPRO + BCRAT (M) overestimated risk in the top decile of risk and IBIS overestimated risk in the second highest decile, while BRCAPRO and BCRAT underestimated risk in other deciles. All models except BRCAPRO overpredicted risk by a certain amount in the second-smallest decile, suggesting that the source of this pattern may reside in specific characteristics of the distribution of covariates other than family history.

The AUCs were 0.68 (95% CI 0.63–0.72) for the combination models, 0.67 for IBIS (95% CI 0.62–0.71), 0.66 for BCRAT (95% CI 0.61–0.71), and 0.65 (95% CI 0.60–0.69) for BRCAPRO. IBIS had the highest SNB (SNB = 0.28, 95% CI 0.16–0.38), followed by BRCAPRO + BCRAT (M) (SNB = 0.24, 95% CI 0.13–0.35) and the ensemble models (SNP = 0.24, 95% CI 0.12–0.34 for (E2), SNB = 0.23 and SNP = 0.23, 95% CI 0.10–0.33 for (E)). All models performed similarly with respect to the Brier score and logarithmic score. Across 1000 bootstrap replicates, BRCAPRO + BCRAT (M) outperformed BRCAPRO and BCRAT with respect to all performance measures except the Brier score in the majority of the replicates. Each ensemble model outperformed BRCAPRO and BCRAT with respect to all metrics except O/E in the majority of the replicates. Furthermore, each of the three combination models outperformed IBIS with respect to AUC, Brier score, and logarithmic score in the majority of the replicates, although IBIS had better calibration and SNB in most replicates.

Among counselees who met the NCCN criteria for further genetic risk evaluation (Table 4), the combination models and IBIS had higher AUCs and SNBs than BRCAPRO and BCRAT. BRCAPRO + BCRAT (M) and IBIS overestimated risk while all other models except BCRAT underestimated risk. Among counselees who did not meet the NCCN criteria, all models underestimated risk. BRCAPRO had a slightly lower AUC than the other models, and IBIS had the highest SNB.

#### 3.2.2. Time-to-Event Outcome

The performance measures based on the time-to-event outcome are shown in Table A5 (overall performance) and Table A6 (performance stratified by family history) in Appendix D. BRCAPRO + BCRAT (E) was excluded from the analyses because it only provides five-year risks. We did not calculate the logarithmic score for BRCAPRO + BCRAT (E2) or IBIS because the cause-specific distribution for competing mortality, which is needed to calculate the likelihood, is not explicitly modelled by BRCAPRO + BCRAT (E2) and is not available as an output from the IBIS software.

Overall, the relative performance of the models for the time-to-event outcome was similar to the relative performance for the five-year outcome; however, among counselees meeting the NCCN criteria BCRAT had worse discriminatory accuracy for the time-to-event outcome than the other models. O/E and discriminatory accuracy did not change substantially compared to Table 3 and Table 4, and differences in Brier score and logarithmic score across models were small. BRCAPRO + BCRAT (M) and BRCAPRO + BCRAT (E2) performed similarly to BCRAT and IBIS overall (Table A5), and had higher C-statistics than BCRAT in the subset of counselees meeting the NCCN criteria (Table A6). The combination models had similar or better calibration and discrimination compared to BRCAPRO overall as well as within each family history stratum.

## 4. Discussion

Model combination is a way to systematically integrate information from different models to achieve more comprehensive and accurate risk assessment. By leveraging existing models instead of building new ones from the ground up, it is possible to save considerable time and effort, remove barriers to adoption if the ingredient models are well-accepted, and, in the case of ensemble approaches such as stacking, permit reuse of proprietary models as long as predictions can be obtained even when the full model is not available.

We used model combination to expand a widely used Mendelian model, BRCAPRO, to include non-genetic risk factors from a widely used non-Mendelian model. We considered two approaches, namely, penetrance modification and ensemble learning. The penetrance modification model BRCAPRO + BCRAT (M) and the ensemble models BRCAPRO + BCRAT (E) for binary outcomes and BRCAPRO + BCRAT (E2) for time-to-event outcomes all achieved accuracy gains over BRCAPRO and BCRAT in simulations and data from the CGN, showing the value of model combination. Augmenting the family history input to BRCAPRO with non-genetic risk factors and augmenting the BCRAT risk factors with more detailed family history information both led to improvements, and could provide a better tool to users of either BRCAPRO or BCRAT as well as to clinicians who use both and then qualitatively integrate the two estimates for their patients.

In data simulated under the penetrance modification model, all combination models outperformed BRCAPRO and BCRAT with respect to calibration, discrimination, net benefit, and prediction accuracy based on the Brier score and logarithmic score. In the CGN cohort, we additionally validated IBIS, an existing model that combines detailed family history with non-genetic risk factors. The penetrance modification model achieved comparable performance to IBIS overall, outperforming BRCAPRO with respect to each performance measure and outperforming BCRAT with respect to each measure except the Brier score in a large majority of bootstrap replicates. The ensemble models outperformed BRCAPRO with respect to each measure and BCRAT with respect each measure other than calibration, showing worse overall calibration than the penetrance modification model. In the subset of women with a strong family history of breast/ovarian cancer based on NCCN criteria, all combination models showed notable improvements in discrimination and net benefit over BRCAPRO and BCRAT.

While the AUC gains of 2–3% achieved by the combination models over BRCAPRO and BCRAT in the entire CGN cohort may not change the clinical implications for most individuals, the net benefit measure, which aims to quantify the expected benefits versus harms of treatment and has been used to assess the clinical utility of numerous models in cancer research and other fields [45,46], provides additional information on the value of the combination models. The 5–7% gain in standardized net benefit (based on a high-risk threshold of 1.67%, the FDA threshold for chemoprevention eligibility) indicates that there is a subset of individuals who would be accurately classified by the combination models and inaccurately classified by BRCAPRO/BCRAT. The combination models would be highly valuable for this subgroup. In particular, among women with a strong family history of cancer, the combination models achieved a 5% improvement in AUC and 10–14% improvement in standardized net benefit over BRCAPRO and BCRAT. In this subgroup, early screening and prevention measures can substantially reduce cancer risk and mortality [56]. Figure 3 highlights the substantial number of counselees for whom differences in model predictions are large enough to warrant different discussions about the expected benefits of preventative options.

BCRAT performed well in the entire CGN cohort, although it had less discrimination and net benefit than the combination models and IBIS among women with a strong family history, highlighting the importance of collecting detailed family history information for higher-risk subgroups [37]. Furthermore, BCRAT is not designed to calculate risk for known BRCA1/2 carriers, while the other four models all take into account genetic testing results (which the ensemble models do indirectly through BRCAPRO risk prediction).

The additional inputs in the combination models compared to BRCAPRO or BCRAT alone require additional data collection effort; however, models integrating detailed family history and non-genetic risk factors are already used in clinical practice (IBIS and BOADICEA), and these models use similar inputs compared to our combination models. Furthermore, missing values are allowed in our combination models, which can help to reduce the burden of data collection in clinical practice.

A limitation of our study arises from missing information on BCRAT and IBIS risk factors in the CGN dataset (atypical hyperplasia, age at menarche, and for IBIS, breast density, hormone replacement therapy, and polygenic risk scores). This could potentially have affected the discrimination of BCRAT, IBIS, and the combination models as well as their relative ranking. To mitigate this concern, we observe that the models retain relatively good discrimination that is comparable to previous studies [13,16]. Furthermore, the CGN did not collect genetic testing information for non-counselees, which could considerably improve the discrimination of BRCAPRO, IBIS, and the combination models [57].

Another limitation is that the CGN and NWH cohorts are predominantly non-Hispanic white (>80%), with <5% Black and <10% Hispanic. The combination models evaluated here do not overcome the potential limitations of BRCAPRO and BCRAT in predicting risk among racial/ethnic minorities, at least as evaluated in the current datasets. While one recent study evaluated BRCAPRO, BCRAT, IBIS, and BRCAPRO + BCRAT (M) in a diverse US cohort and found no significant difference in model performance between Black and White women [58], BCRAT has shown only modest discrimination in other validation studies focusing on African-American, Asian-American, and US-born Hispanic women [5,6,7], and its performance has varied in non-US populations [59]. Moreover, other validation studies of BRCAPRO in minority population have shown its clinical utility for predicting carrier probabilities [60,61], although its performance for predicting future risk has not been evaluated. Future work should evaluate the combination models among more diverse populations.

A further challenge is that the NWH and CGN cohorts are not representative of the same clinical population. The NWH cohort is a lower-risk cohort; as seen in Table 1, it has a lower proportion of women with a first-degree family history of breast cancer. In addition, the family history information available for the NWH cohort is less detailed than that for the CGN cohort. While this allows us to assess the robustness of a model to heterogeneity across different datasets, it creates a challenge for the ensemble models, which are trained in NWH and validated in CGN, compared to the penetrance modification model, which leverages literature estimates based on populations potentially more similar to CGN. We used importance weighting to address this limitation; however, this approach remains an approximation and relies on accurate estimation of the probability distributions of risk factors in the training and target populations. The performance of the ensemble approach could potentially be improved by training on data that are more representative of the validation data. This is supported by the fact that the ensemble models were well-calibrated in the simulations, where the training and validation datasets were both generated under BRCAPRO + BCRAT (M).

The two combination approaches considered here each have their strengths and limitations. Ensembling via logistic regression calibrates the model to the training data. The penetrance modification model, on the other hand, relies on previously published estimates of prevalence, penetrance, and relative hazards. These features can be a strength or a limitation depending on how well the training dataset represents the target population. While ensembling can sometimes be successful without training weights (for example, predictions can be combined using a simple average), in our approach we used stacking to train weights on individual-level data. This opens opportunities for recalibration to improve suitability for a population different from the training population.

A disadvantage of estimating ensemble weights via logistic regression is the need to estimate the censoring distribution when there are counselees in the training dataset who are censored before the last time point of interest. An advantage of ensembling is its greater flexibility compared to the penetrance modification approach. Ensembling can easily handle any number of models that can be of any form. Including additional risk factors is straightforward. The penetrance modification model requires more assumptions, as it specifically combines a model based on Mendelian inheritance with a relative hazard model. Additional risk modifiers can be incorporated as new relative hazard estimates become available; however, it is important to properly scale the relative hazards to be compatible with the hazard functions they are meant to modify and to consider whether the effects of the risk modifiers differ by carrier status. BRCAPRO + BCRAT (M) currently modifies only the non-carrier hazard function, for which it uses the BCRAT relative hazard. Future work is needed to evaluate the inclusion of modifiers of the carrier hazard functions. One more advantage of ensembling is that after the model is trained it only requires the predictions from the models being combined (along with any additional risk factors that are explicitly included in the ensemble model), and does not require the raw inputs, which are potentially less accessible than the predictions.

BRCAPRO + BCRAT (M) has been externally validated in two average-risk cohorts: a large US mammography screening cohort of over 120,000 women [58] and an Australian cohort of over 7000 women [14]. In the US cohort, BRCAPRO + BCRAT (M) performed better than BRCAPRO, especially among women with a family history of cancer, and performed similarly to BCRAT. In the Australian cohort, BRCAPRO + BCRAT (M) performed similarly to BRCAPRO and worse than BCRAT. To investigate whether these results may have been driven by cohort characteristics, we simulated data to mimic the characteristics of the Australian cohort, as the original data are not publicly available. In these additional simulations, we were unable to reproduce the results reported in the Australian study. Rather than seeing similar performance of BRCAPRO and BRCAPRO + BCRAT (M), as shown in [14], in our simulations BRCAPRO + BCRAT (M) outperformed BRCAPRO alone, which is consistent with the results from the US cohort validation [58] (Appendix E).

While at least two breast cancer risk prediction models integrating detailed family history and non-genetic risk factors, namely, IBIS and BOADICEA, are well-validated and widely available, our work contributes to the improvement of clinical risk prediction in the following ways: (1) we have developed an expanded version of BRCAPRO that can easily be integrated in clinical and research settings where BRCAPRO is currently used, and (2) the model combination methodology proposed in this paper is applicable to a wide range of prediction problems. BRCAPRO is currently part of seven widely used and commercially available clinical tools. These have thousands of users, not all of whom have access to IBIS and BOADICEA. The proposed BRCAPRO + BCRAT (M) model has been added to the software implementation of BRCAPRO, which can facilitate its integration in clinical and research settings where BRCAPRO is used. Two useful features of BRCAPRO are the free availability of its software and code (through the BayesMendel R package) for research purposes and the option for users to specify their own values for the model parameters (through the brcaparams function in the BayesMendel R package), including modifications to the allele frequencies and penetrance instead of using the default published estimates. This provides researchers with the flexibility to tailor the model to different populations as well as an easy way to check the sensitivity of risk predictions to differing sets of parameter values. While risk calculation tools for IBIS and BOADICEA are freely available, to the best of our knowledge the code is not easily accessible and there is no option to customize and implement user-specified parameters. Furthermore, the model combination approaches we propose can be applied to many other prediction problems. Ensemble learning can be used to combine an arbitrary number of models of any type. Penetrance modification is a way to incorporate risk factors beyond family history into complex Mendelian models, including the recently developed multi-gene and multi-syndrome model PanelPro [15,62], which can incorporate an arbitrary number of genes and cancer types and can potentially provide greater clinical benefit than single-syndrome models such as BRCAPRO.

## 5. Conclusions

In summary, we have demonstrated the feasibility of two complementary and methodologically different approaches to integrating the popular breast cancer risk assessment models BRCAPRO and BCRAT. Both combination models show accuracy improvements over the existing models, and can potentially prove effective in clinical application. In the CGN validation, BRCAPRO + BCRAT (M) achieved comparable performance to the IBIS model by leveraging the strengths of BRCAPRO and BCRAT. Additional validation of BRCAPRO + BCRAT (M) in independent prospective studies, ideally with larger and more diverse cohorts and more complete covariate information, would provide further support for clinical adoption.

## Figures and Tables

**Figure 1 cancers-15-01090-f001:**
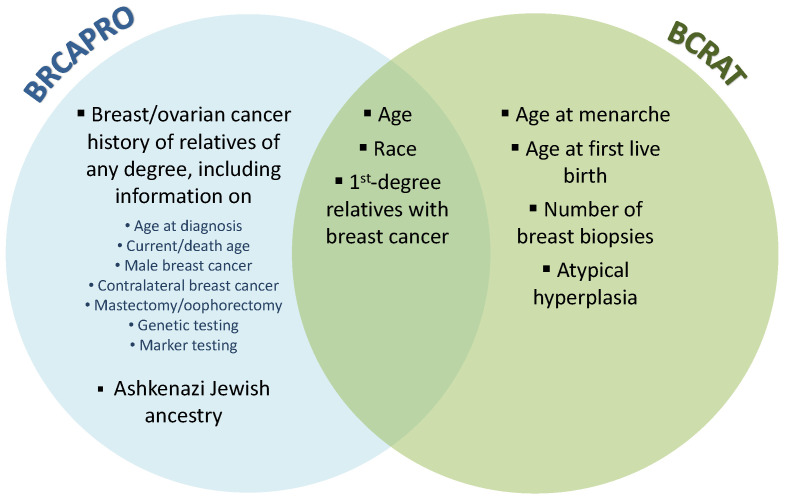
Inputs to BRCAPRO and BCRAT and their overlap.

**Figure 2 cancers-15-01090-f002:**
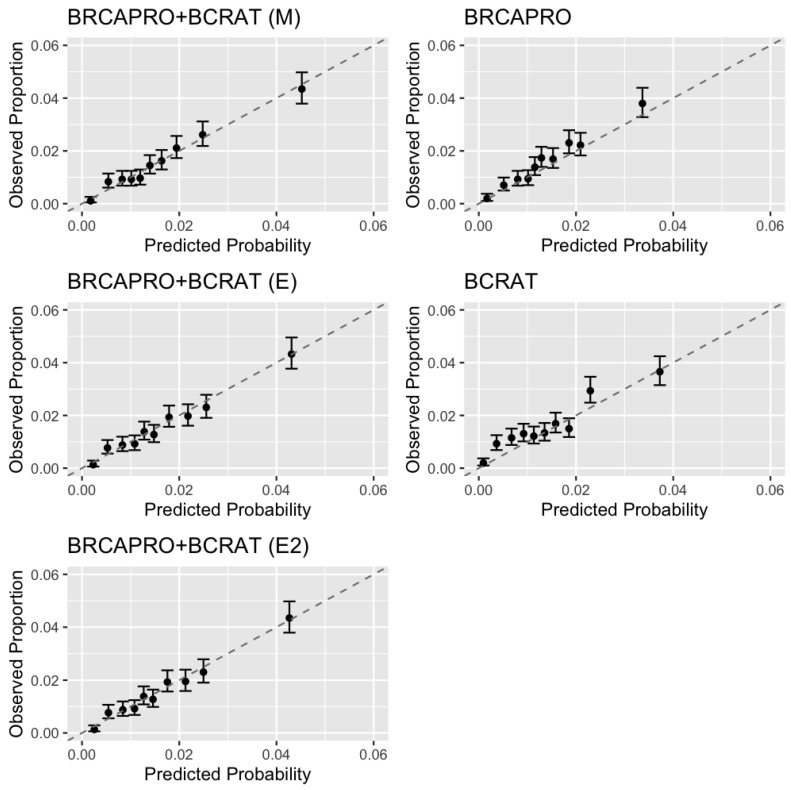
Calibration plots by decile of risk for five-year predictions in a simulated dataset with 45,557 counselees (724 cases). For each model, we grouped individuals by decile of risk and plotted the observed proportion of women who developed cancer (with 95% Wilson CI) versus the predicted probability (sum of risk predictions) within each decile.

**Figure 3 cancers-15-01090-f003:**
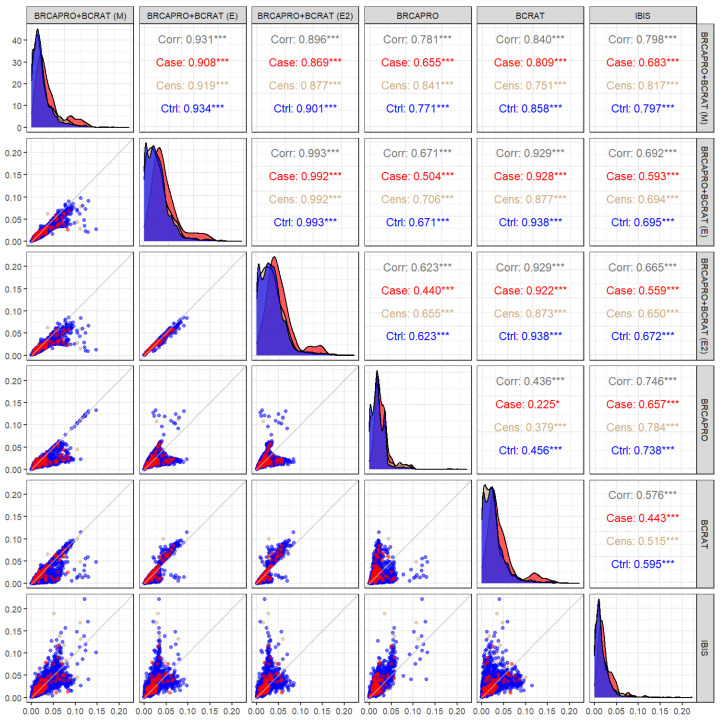
Scatter plots, density plots, and pairwise correlations of five-year predictions in the CGN data. Red corresponds to data from cases, blue corresponds to data from controls (individuals who did not develop breast cancer within five years), and beige corresponds to data from counselees censored before five years. Corr: Pearson correlation, cens: counselees censored before five years, ctrl: controls. The lower diagonal panels show scatter plots of the predictions from each pair of models. For example, the panel in the second row, first column has predictions from BRCAPRO + BCRAT (E) on the x-axis and predictions from BRCAPRO + BCRAT (M) on the y-axis. The diagonal panels show density plots of the predictions from each model, stratified by case-control status. For example, the panel in the first row, first column shows the distribution of predictions from BRCAPRO + BCRAT (M). The upper diagonal panels show the Pearson correlations between predictions from each pair of models. For example, the first row, second column shows the overall correlation, correlation among cases, correlation among censored counselees, and correlation among controls for predictions from BRCAPRO + BCRAT (M) and BRCAPRO + BCRAT (E). *** = *p* < 0.001.

**Figure 4 cancers-15-01090-f004:**
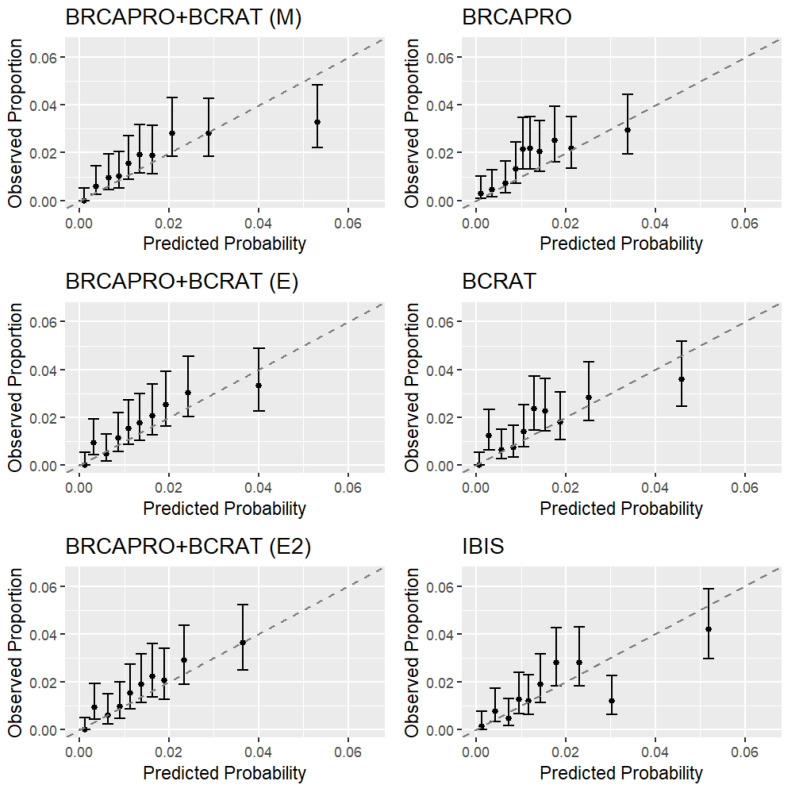
Calibration plots by decile of risk for five-year predictions in CGN. For each model, we grouped individuals by decile of risk and plotted the observed proportion of women who developed cancer (with 95% Wilson CI) versus predicted probability (sum of risk predictions) within each decile. In computing the observed proportions, the inverse probabilities of the censoring weights were used to account for censoring.

**Table 1 cancers-15-01090-t001:** Main characteristics of the CGN and NWH cohorts.

Variable	Category	CGN	NWH
N		7314	37,881
Age (median [IQR])		47 [38, 57]	49 [43, 58]
Race (%)	White	6104 (83.5)	30,758 (81.2)
	Black	257 (3.5)	479 (1.3)
	Hispanic	694 (9.5)	548 (1.4)
	Asian	160 (2.2)	1228 (3.2)
	Native American	29 (0.4)	25 (0.1)
	Unknown	70 (1.0)	4843 (12.8)
Affected 1st-degree Relatives (%)	0	4171 (57.0)	32,197 (85.0)
	1	2496 (34.1)	5277 (13.9)
	2+	647 (8.8)	407 (1.1)
Follow-up Time (median [IQR])		7.3 [6.0, 8.3]	6.7 [6.3, 7.2]
Censored < 5 Years (%)		934 (12.8)	0 (0.0)
Cases (%)		159 (2.2)	714 (1.9)
5-year Cases (%)		112 (1.5)	495 (1.3)

**Table 2 cancers-15-01090-t002:** Five-year performance on a simulated dataset with 45,557 probands (717 cases). B+B: BRCAPRO + BCRAT. ΔBS: % relative improvement in Brier Score compared to BRCAPRO. ΔLS: % relative improvement in logarithmic score compared to BRCAPRO. The “Comparisons Across Bootstrap Replicates” section shows pairwise comparisons involving the combination models across 1000 bootstrap replicates of the validation dataset; the row for A>B shows the proportion of bootstrap replicates where model A outperformed model B with respect to each metric. Proportions >0.5 are highlighted in blue (with darker shades of blue for higher proportions) and proportions ≤0.5 are highlighted in red (with darker shades of red for lower proportions).

	O/E	AUC	SNB	ΔBS	ΔLS
**Performance Metrics**					
B+B (M)	1.01 (0.94, 1.08)	0.69 (0.67, 0.71)	0.26 (0.21, 0.30)	0.25 (0.09, 0.41)	1.12 (0.60, 1.66)
B+B (E)	0.98 (0.91, 1.04)	0.68 (0.67, 0.70)	0.25 (0.20, 0.29)	0.12 (−0.02, 0.25)	0.63 (0.18, 1.05)
B+B (E2)	0.99 (0.92, 1.05)	0.68 (0.67, 0.70)	0.24 (0.19, 0.29)	0.16 (0.01, 0.29)	0.65 (0.21, 1.07)
BRCAPRO	1.15 (1.07, 1.23)	0.67 (0.65, 0.69)	0.21 (0.17, 0.25)	0.00 (0.00, 0.00)	0.00 (0.00, 0.00)
BCRAT	1.14 (1.05, 1.21)	0.66 (0.64, 0.68)	0.20 (0.15, 0.24)	−0.21 (−0.46, 0.06)	−1.15 (−2.10, −0.16)
**Comparisons Across Bootstrap Replicates**					
B+B(M)>B+B(E)	0.570	0.994	0.725	0.999	1.000
B+B(M)>B+B(E2)	0.504	0.995	0.795	0.972	1.000
B+B(M)>BRCAPRO	0.978	0.999	0.998	0.998	1.000
B+B(M)>BCRAT	0.962	1.000	1.000	1.000	1.000
B+B(E)>B+B(E2)	0.312	0.919	0.808	0.011	0.133
B+B(E)>BRCAPRO	0.944	0.999	1.000	0.961	0.998
B+B(E)>BCRAT	0.924	1.000	0.999	0.994	1.000
B+B(E2)>BRCAPRO	0.953	0.999	0.995	0.979	0.999
B+B(E2)>BCRAT	0.938	1.000	0.998	0.996	1.000

**Table 3 cancers-15-01090-t003:** Five-year performance in the entire CGN cohort. B+B: BRCAPRO + BCRAT. ΔBS: % relative improvement in Brier Score compared to BRCAPRO. ΔLS: % relative improvement in logarithmic score compared to BRCAPRO. The “Comparisons Across Bootstrap Replicates” section shows pairwise comparisons involving the combination models across 1000 bootstrap replicates of the validation dataset; the row for A>B shows the proportion of bootstrap replicates where model A outperformed model B with respect to each metric. Proportions >0.5 are highlighted in blue (with darker shades of blue for higher proportions) and proportions ≤0.5 are highlighted in red (with darker shades of red for lower proportions).

	O/E	AUC	SNB	ΔBS	ΔLS
**Performance Metrics**					
B+B (M)	1.03 (0.84, 1.23)	0.68 (0.63, 0.72)	0.24 (0.13, 0.35)	0.21 (−0.42, 0.84)	1.61 (−0.49, 3.59)
B+B (E)	1.17 (0.96, 1.40)	0.68 (0.63, 0.72)	0.23 (0.10, 0.33)	0.38 (−0.10, 0.84)	1.74 (−0.06, 3.41)
B+B (E2)	1.20 (0.98, 1.43)	0.68 (0.63, 0.72)	0.24 (0.12, 0.34)	0.40 (−0.07, 0.85)	1.81 (0.06, 3.48)
BRCAPRO	1.30 (1.07, 1.56)	0.65 (0.60, 0.69)	0.17 (0.05, 0.25)	0.00 (0.00, 0.00)	0.00 (0.00, 0.00)
BCRAT	1.15 (0.94, 1.37)	0.66 (0.61, 0.71)	0.18 (0.07, 0.28)	0.22 (−0.51, 0.85)	0.88 (−2.03, 3.60)
IBIS	0.98 (0.80, 1.17)	0.67 (0.62, 0.71)	0.28 (0.16, 0.38)	−0.05 (−0.69, 0.52)	0.89 (−1.43, 3.07)
**Comparisons Across Bootstrap Replicates**					
B+B(M)>B+B(E)	0.841	0.421	0.702	0.208	0.412
B+B(M)>B+B(E2)	0.874	0.382	0.534	0.225	0.381
B+B(M)>BRCAPRO	0.943	0.920	0.933	0.727	0.934
B+B(M)>BCRAT	0.811	0.763	0.982	0.503	0.783
B+B(M)>IBIS	0.475	0.709	0.186	0.823	0.774
B+B(E)>B+B(E2)	0.963	0.310	0.001	0.288	0.266
B+B(E)>BRCAPRO	0.983	0.965	0.913	0.936	0.970
B+B(E)>BCRAT	0.066	0.844	0.870	0.895	0.907
B+B(E)>IBIS	0.214	0.717	0.119	0.919	0.806
B+B(E2)>BRCAPRO	0.987	0.955	0.950	0.953	0.980
B+B(E2)>BCRAT	0.047	0.870	0.937	0.895	0.938
B+B(E2)>IBIS	0.181	0.739	0.189	0.918	0.823

**Table 4 cancers-15-01090-t004:** Five-year performance in the CGN cohort stratified by family history (whether or not the proband met the NCCN criteria for further genetic risk evaluation [55]; in applying the criteria, we only used information on breast and ovarian cancer diagnoses in relatives). B+B: BRCAPRO + BCRAT. ΔBS: % relative improvement in Brier Score compared to BRCAPRO. The “Comparisons Across Bootstrap Replicates” section shows pairwise comparisons involving the combination models across 1000 bootstrap replicates of the validation dataset; the row for A>B shows the proportion of bootstrap replicates where model A outperformed model B with respect to each metric. Proportions >0.5 are highlighted in blue (with darker shades of blue for higher proportions) and proportions ≤0.5 are highlighted in red (with darker shades of red for lower proportions).

	O/E	AUC	SNB	ΔBS	ΔLS
**Strong Family History (34 cases)**					
**Performance Metrics**					
B+B (M)	0.81 (0.55, 1.09)	0.71 (0.63, 0.79)	0.44 (0.18, 0.59)	0.75 (−1.20, 2.14)	3.62 (−2.27, 8.06)
B+B (E)	1.07 (0.74, 1.44)	0.71 (0.62, 0.79)	0.41 (0.17, 0.57)	1.19 (−0.16, 2.54)	4.15 (−0.77, 8.39)
B+B (E2)	1.14 (0.78, 1.53)	0.71 (0.62, 0.79)	0.41 (0.17, 0.57)	1.11 (−0.19, 2.49)	3.90 (−1.07, 8.29)
BRCAPRO	1.32 (0.91, 1.79)	0.66 (0.58, 0.74)	0.30 (0.08, 0.47)	0.00 (0.00, 0.00)	0.00 (0.00, 0.00)
BCRAT	1.03 (0.71, 1.39)	0.66 (0.56, 0.75)	0.31 (0.06, 0.47)	0.77 (−1.36, 2.65)	1.55 (−6.64, 8.24)
IBIS	0.74 (0.50, 1.00)	0.69 (0.60, 0.77)	0.41 (0.14, 0.57)	−0.15 (−2.52, 1.46)	1.12 (−5.56, 5.92)
**Comparisons Across Bootstrap Replicates**					
B+B(M)>B+B(E)	0.339	0.593	0.634	0.192	0.318
B+B(M)>B+B(E2)	0.409	0.584	0.633	0.272	0.408
B+B(M)>BRCAPRO	0.615	0.931	0.917	0.794	0.892
B+B(M)>BCRAT	0.278	0.906	0.965	0.463	0.778
B+B(M)>IBIS	0.951	0.706	0.718	0.880	0.920
B+B(E)>B+B(E2)	0.704	0.521	0.423	0.699	0.718
B+B(E)>BRCAPRO	0.841	0.853	0.905	0.956	0.949
B+B(E)>BCRAT	0.410	0.945	0.879	0.856	0.941
B+B(E)>IBIS	0.751	0.669	0.532	0.919	0.897
B+B(E2)>BRCAPRO	0.876	0.834	0.903	0.949	0.933
B+B(E2)>BCRAT	0.335	0.954	0.877	0.772	0.915
B+B(E2)>IBIS	0.664	0.652	0.541	0.901	0.874
**Less Family History (78 cases)**					
**Performance Metrics**					
B+B (M)	1.16 (0.94, 1.44)	0.65 (0.60, 0.71)	0.16 (0.02, 0.28)	−0.02 (−0.41, 0.37)	0.80 (−0.92, 2.47)
B+B (E)	1.21 (0.98, 1.49)	0.66 (0.61, 0.71)	0.14 (0.01, 0.27)	0.03 (−0.22, 0.26)	0.80 (−0.54, 2.08)
B+B (E2)	1.22 (0.98, 1.50)	0.66 (0.61, 0.71)	0.16 (0.03, 0.29)	0.10 (−0.16, 0.33)	1.02 (−0.29, 2.34)
BRCAPRO	1.28 (1.04, 1.59)	0.63 (0.58, 0.68)	0.10 (−0.02, 0.21)	0.00 (0.00, 0.00)	0.00 (0.00, 0.00)
BCRAT	1.20 (0.97, 1.48)	0.66 (0.61, 0.71)	0.12 (−0.00, 0.24)	−0.01 (−0.41, 0.35)	0.68 (−1.55, 2.92)
IBIS	1.14 (0.92, 1.41)	0.66 (0.61, 0.71)	0.22 (0.09, 0.34)	−0.01 (−0.38, 0.39)	0.81 (−1.38, 3.11)
**Comparisons Across Bootstrap Replicates**					
B+B(M)>B+B(E)	0.937	0.275	0.623	0.333	0.490
B+B(M)>B+B(E2)	0.940	0.196	0.429	0.233	0.329
B+B(M)>BRCAPRO	0.964	0.875	0.822	0.432	0.813
B+B(M)>BCRAT	0.934	0.223	0.900	0.424	0.542
B+B(M)>IBIS	0.101	0.396	0.121	0.429	0.476
B+B(E)>B+B(E2)	0.961	0.126	0.000	0.001	0.008
B+B(E)>BRCAPRO	0.979	0.958	0.806	0.630	0.890
B+B(E)>BCRAT	0.055	0.385	0.687	0.715	0.588
B+B(E)>IBIS	0.075	0.498	0.106	0.594	0.477
B+B(E2)>BRCAPRO	0.982	0.955	0.875	0.821	0.923
B+B(E2)>BCRAT	0.043	0.446	0.875	0.939	0.742
B+B(E2)>IBIS	0.069	0.532	0.168	0.744	0.579

## Data Availability

De-identified data that support the findings of this study are available on request. The data are not publicly available due to privacy restrictions. The simulation code is available at https://github.com/zoeguan/brcapro_bcrat_combination (accessed on 20 July 2020).

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
