# Peer review of "Combining Breast Cancer Risk Prediction Models"

_cancers, 2023, doi:10.3390/cancers15041090_

Round 1

Reviewer 1 Report

This is an interesting approach to a common problem.  My only question is the value add with the widespread availability of IBIS.  

Reviewer 2 Report

The Author used simulated data and data from two sources to demonstrate that the combination models can achieve performance gains over BRCAPRO and BCRAT. The methods used are well-considered and designed. The results are also presented nicely with visualization. 

Major concerns: the bootstrapped "performance gain" just ensured the increase of AUC for 1%-2% over commonly used models in practice could be true. This gain margin does not have a clinical impact. The combined model would require more risk factors input collected with more effort, compared with either BRCAPRO or BCRAT, but improve prediction very little.

The paper used an indirect way to answer whether combining family history input with non-genetic risk factors could lead to prediction improvements and could provide a better tool. It is understandable that major limitations are coming from data availability. It is also agreeable that it is more efficient to develop a new model this way. However, the author claimed the significance of the findings for both users of BRCAPRO and BCRAT, and clinicians to integrate the two estimates for their patients. This would not be true. The user and clinician would not be interested to use such a model at all, as 1) a performance gain of <3-5% gives no risk stratification or clinical decision support benefit 2) similar models with both genetic and non-genetic input already exist and performs comparably 3) significant more data collection efforts needed to use such a model. This manuscript is interesting for researchers into disease predictive modeling who have a curiosity about such methodology (Ensemble Models).

Minor concerns:

1 the formulas in 2.3 can be explained with words and moved into the appendix together with the original models to make it easier to read

2 Please make sure that models 6 formula and A16-18 are correct

3 Figures 3 and 4 did not provide easily understandable visualization 

Reviewer 3 Report

The authors present the results of an analysis evaluating a combination model of BRCAT and BRCAPRO for breast cancer risk prediction using both clinical and genetic/hereditary factors. Overall, this is a well-conducted and well-written study. I suggest that they add the following limitation to their discussion:

1. The populations of the CGN and NWH are predominantly non-Hispanic white (>80%), with <5% Black and <10% Hispanic. A major limitation of existing risk models, including BRCAT, is that they are inaccurate among racial/ethnic minorities. The combination models evaluated here do not overcome that potential limitation, at least as evaluated in the current datasets. Future work will have to evaluate the combination models among more diverse populations.

Round 2

Reviewer 2 Report

The authors addressed my comments well. The discussion has also been improved in revision.  I have no further comments.